# Development of Monoclonal Antibody Specific to Foot-and-Mouth Disease Virus Type A for Serodiagnosis

**DOI:** 10.3390/pathogens8040301

**Published:** 2019-12-17

**Authors:** Quyen Thi Nguyen, Jihyun Yang, Jae-Won Byun, Hyun Mi Pyo, Mi-Young Park, Bok Kyung Ku, Jinju Nah, Soyoon Ryoo, Sung-Hwan Wee, Kang-Seuk Choi, Haryoung Poo

**Affiliations:** 1Infectious Disease Research Center, Korea Research Institute of Bioscience and Biotechnology (KRIBB), Daejeon 34141, Korea; quyenbio@kribb.re.kr (Q.T.N.); jhyang@kribb.re.kr (J.Y.); 2Department of Biosystems and Bioengineering, KRIBB School of Biotechnology, University of Science and Technology, Daejeon 34113, Korea; 3Foot-and-Mouth Disease Division, Animal and Plant Quarantine Agency, Gyongsangbuk-do 39660, Korea; jaewon8911@korea.kr (J.-W.B.); hmpyo@korea.kr (H.M.P.); parkmy71@korea.kr (M.-Y.P.); kubk@korea.kr (B.K.K.); nahjj75@korea.kr (J.N.); soyooni@korea.kr (S.R.); wsh2010@korea.kr (S.-H.W.); kchoi0608@korea.kr (K.-S.C.)

**Keywords:** foot-and-mouth disease, type A, structural protein, monoclonal antibody, solid-phase competitive ELISA, sensitivity

## Abstract

Foot-and-mouth disease (FMD) is a highly contagious and economically devastating disease affecting cloven-hoofed livestock worldwide. FMD virus (FMDV) type A is one of the most common causes of FMD outbreaks among the seven FMDV serotypes, and its serological diagnosis is therefore important to confirm FMDV type A infection and to determine FMD vaccine efficacy. Here, we generated monoclonal antibodies (mAbs) specific to FMDV type A via hybridoma systems using an inactivated FMDV type A (A22/Iraq/1964) and found 4 monoclones (#29, #106, #108, and #109) with high binding reactivity to FMDV type A among 594 primary clones. In particular, the #106 mAb had a higher binding reactivity to the inactivated FMDV type A than the other mAbs and a commercial mAb. Moreover, the #106 mAb showed no cross-reactivity to inactivated FMDV type South African territories 1, 2, and 3, and low reactivity to inactivated FMDV type O (O_1_ Manisa). Importantly, the solid-phase competitive ELISA (SPCE) using horseradish peroxidase (HRP)-conjugated #106 mAb detected FMDV type A-specific Abs in sera from FMD type A-vaccinated cattle more effectively than a commercial SPCE. These results suggest that the newly developed FMDV type A-specific mAb might be useful for diagnostic approaches for detecting Abs against FMDV type A.

## 1. Introduction

Foot-and-mouth disease (FMD) is a highly contagious disease in cloven-hoofed animals and is caused by infection with an FMD virus (FMDV), a member of the genus *Aphthovirus* in the family *Picornaviridae* [1]. Because FMDV can rapidly spread between susceptible animals, the disease is listed as one of the most important animal diseases by the World Organization for Animal Health. FMD outbreaks result in a devastating impact on economies due to constraints on the international trade of livestock and animal products [2,3,4]. FMDV exists in seven distinct serotypes comprising O, Asia 1, A, C, and South African territory (SAT) 1, 2, and 3 [5,6]. FMDV type A is one of the most widespread FMDV serotypes worldwide, and FMD type A outbreaks occur in many countries, including South Korea [7]. Thus, an inactivated FMD vaccine using a predominant FMDV type A strain, A22/Iraq/1964, has been widely used for preventing FMDV type A infections [8,9,10].

Recently, various diagnostic approaches, including the virus neutralization test (VNT), liquid-phase blocking ELISA (LPBE), and solid-phase competitive ELISA (SPCE), have been internationally accepted for detecting FMDV-specific antibodies (Abs) after vaccination and infection [11]. VNT is considered the gold standard for detecting Abs to structural proteins (SPs) of FMDV, but it has several limitations, such as requiring restrictive biocontainment facility, being time consuming, and having high costs. In addition, the VNT is more prone to variability than ELISA-based tests because of the use of various primary cells and cell lines with different sensitivities. Due to its ease of use, LPBE has been applied as the routine FMDV screening method, but it also has several drawbacks, including a lack of antigen stability and false positive reactions [12,13]. SPCE is an assay based on a competition between sera Abs and antigen-specific monoclonal Ab (mAb) to bind to antigens and has been developed for detecting FMD Abs [14,15]. Notably, SPCE has been reported to have a higher specificity than the LPBE for detecting Abs to SPs of the FMDV [16].

Since the first reported FMDV type A outbreak in South Korea in 2010 [17], the Korean government adopted a routine vaccination program against FMDV type A. Despite this effort, the outbreak of FMD type A occurred in pigs in 2018 and put animal health authorities on alert. Currently, vaccination is considered the best strategy for controlling FMD outbreaks, and therefore postvaccination serological tests become an important indicator for evaluating preventive immunization programs. SPCE has been adopted as a screening method for evaluating the immune status after FMD vaccination, because VNTs require more time and is more labor-consuming than SPCE. For effective FMD postvaccination monitoring, it is necessary to improve the sensitivity and specificity of antigen-specific mAbs in SPCE.

In this study, we produced 4 mAbs (#29, #106, #108, and #109) against inactivated FMDV type A (A22/Iraq/1964) via hybridoma systems. The #106 mAb showed a higher binding reactivity to the inactivated FMDV type A than those of the other mAbs and a commercial mAb. In addition, the #106 mAb had no cross-reactivity against inactivated FMDV types SAT 1, 2, and 3 as well as low cross-reactivity to an inactivated FMDV type O (O_1_ Manisa). Importantly, the SPCE using a horseradish peroxidase (HRP)-conjugated #106 mAb more effectively detected FMDV type A-specific Abs in the sera from FMDV type A-vaccinated cattle compared to a commercial SPCE. These findings suggest that the newly developed mAb might be useful for the serodiagnosis for postvaccination of FMDV type A. 

## 2. Results

### 2.1. Production of Anti-FMDV Type A mAbs 

To generate anti-FMDV type A mAbs, we immunized the footpads of BALB/c mice with inactivated FMDV type A (A22/Iraq/1964) mixed with the TiterMax Gold adjuvant on days 0, 14, and 28. Serum samples were obtained from the immunized mice two weeks after each immunization. Production of polyclonal Abs specific to FMDV type A was determined in the sera by ELISA. Bovine serum albumin (BSA) was used as a negative control. As shown in Figure 1A, binding reactivity to FMDV type A antigen dramatically increased in the sera obtained after the second and third immunizations. Two weeks after the last immunization, lymphocytes were obtained from a popliteal lymph node of the immunized mice and were then fused with FO myeloma cells to generate Ab-secreting hybridoma cells. Since FMDV type O is widely distributed among seven serotypes and shares 91.2% identity with the amino acid sequence of FMDV type A [18], ELISA was performed in culture supernatants of primary clones using plates coated with inactivated FMDV type O (O_1_ Manisa) and type A (A22/Iraq/1964) to select primary clones secreting Abs specific to FMDV type A. Among 594 primary clones, we found that the Abs secreted from one primary clone, namely #373, was highly bound to the inactivated FMDV type A but weakly bound to an inactivated FMDV type O; there was a 1.9-fold increase in the OD ratio of FMDV type A to type O (Figure 1B). Then, we selected the primary clone and performed cell subcloning followed by ELISA screening. Of the 109 subclones, the mAb secreted from 4 subclones, designated as #29, #106, #108, and #109, had higher binding reactivities to the inactivated FMDV type A but not the type O compared to those of the other subclones (OD ratio of FMDV type A to type O: 2.9 for #29, 3.1 for #106, 3 for #108, and 2.9 for #109) (Figure 1C). The 4 mAbs were purified from culture supernatants by large-scale cultures following purification using a protein G column for further experiments.

### 2.2. Comparison of Binding Reactivity and Characterization of the mAbs Against FMDV Type A

To compare the binding reactivities of the generated mAbs to FMDV type A, we prepared a 2-fold serial dilution of mAbs and performed ELISAs using the diluted mAbs on the inactivated FMDV type A (A22/Iraq/1964)-coated plates. BSA was used as a negative control. As shown in Figure 2A, ELISA showed that the #106 mAb had the highest OD values compared to those of the other mAbs, indicating that the #106 mAb most effectively bound to the inactivated FMDV type A. We also determined the binding efficacies of the mAbs by calculating the half-maximal effective concentration (EC_50_). The EC_50_ of the #106 mAb was 5 μg/mL, which was almost 2-fold lower than those of the #29 (12 μg/mL) and #109 (10 μg/mL) mAbs, and almost 25-fold lower than that of the #108 mAb (124 μg/mL) (Table 1). To further confirm the binding reactivities of the mAbs, ELISA was performed using a consistent amount of the mAbs on plates coated with serially diluted FMDV type A (A22/Iraq/1964). As expected, the #106 mAb showed significantly higher OD values compared to those of other mAbs (*p* < 0.05) when using the inactivated FMDV type A antigen at 2.5, 5, and 10 ug/mL (Figure 2B). That is, the #106 mAb robustly reacted to the inactivated FMDV type A and was selected for further experiments. Additionally, isotypes of the mAbs were determined using an Ab isotyping kit, and all mAbs were IgG2b, kappa (κ)-chain isotypes (Table 1).

Labeling mAbs with various reporter molecules, such as horseradish peroxidase (HRP), is extensively used for immunoassays [19]. Thus, we conjugated the #106 mAb to HRP using an HRP conjugation kit and confirmed the binding reactivity of the HRP-conjugated #106 mAb by ELISAs using serially diluted HRP-conjugated #106 mAb on inactivated FMDV type A (A22/Iraq/1964)-coated plates or a consistent amount of HRP-conjugated #106 mAb on plates coated with serially diluted FMDV type A. As expected, the HRP-conjugated #106 mAb showed higher OD values compared to those of BSA. Notably, the HRP-conjugated #106 mAb had significantly higher OD values than those of an HRP-conjugated commercial mAb named PrioCHECK conjugate (Figure 2C, D). The EC_50_ of the HRP-conjugated #106 mAb was almost 14-fold lower than that of the commercial PrioCHECK conjugate (5 μg/mL for HRP-conjugated #106 mAb versus 69 μg/mL for the PrioCHECK conjugate) (Table 1). These results indicate that the HRP-conjugated #106 mAb has a higher binding reactivity to FMDV type A than the PrioCHECK conjugate.

### 2.3. Specificity of the mAb against FMDV Serotypes

To assess the specificity of the #106 mAb to different FMDV serotypes, we performed ELISAs using serially diluted, purified #106 mAb, HRP-conjugated #106 mAb, or the commercial mAb (PrioCHECK conjugate) on plates coated with inactivated FMDV types O (O_1_ Manisa), SAT 1, SAT 2, or SAT 3. FMD type A (A22/Iraq/1964) was used as a positive control. As shown in Figure 3A,B, ELISAs showed that both purified #106 mAb and HRP-conjugated #106 mAb had significantly 3-fold higher OD values to FMDV type A than those from other FMDV serotypes when using the maximum amounts of mAb (*p* < 0.001). Even though the #106 mAb reacted to the inactivated FMDV type O, the degree of reactions was significantly lower than that of the inactivated FMDV type A (*p* < 0.001). The PrioCHECK conjugate showed no cross-reactivity to FMDV serotypes O and SAT (Figure 3C). We further confirmed the specificity of the #106 mAb by ELISAs using a consistent amount of the purified #106 mAb or HRP-conjugated #106 mAb on plates coated with various concentrations of the FMDV serotypes. Consistently, both purified and HRP-conjugated #106 mAbs showed basal levels of OD values (0.05–0.09 OD: BSA 0.05–0.08 OD) against the inactivated FMDV types SAT 1, 2, and 3, demonstrating that the #106 mAb had no cross-reactivity to FMD type SAT. We observed that both purified and HRP-conjugated #106 mAbs showed cross-reactivity to the inactivated FMDV type O with increasing amounts of mAb or antigen, but the degree of the OD values against inactivated FMDV type O was significantly 3-fold lower than that against the inactivated FMDV type A when using the maximum amount of antigen (*p* < 0.001) (Figure 3D,E). Cross-reactivity to FMDV serotypes O and SAT was not observed by the PrioCHECK conjugate (Figure 3F). Collectively, our results reveal that the #106 mAb has high specificity against FMDV type A but not FMDV type O.

### 2.4. Evaluation of the mAb in SPCE for Detection of SP Abs from FMDV Type A-Vaccinated Cattle

To investigate whether the generated #106 mAb is useful for the detection of SP Abs against FMDV type A in SPCE, we first applied the HRP-conjugated #106 mAb to SPCE using a FMDV type A reference strong positive and negative sera, and the ability of the #106 mAb was compared with the commercial PrioCHECK conjugate. The HRP-conjugated #106 mAb showed a high OD value in the FMDV type A negative sera, which was comparable to the OD value of the PrioCHECK conjugate. In contrast, in the presence of the FMDV type A strong positive sera, both the HRP-conjugated #106 mAb and PrioCHECK conjugate had OD values as low as basal levels. The percentage of inhibition (PI) values for SPCE was also similarly observed using the HRP-conjugated #106 mAb and PrioCHECK conjugate (80 ± 0.45% and 77 ± 0.16%, respectively) (Figure 4A), suggesting that the generated mAb can be applied to detect the presence of FMDV type A-specific SP Abs in the sample from susceptible animals.

We finally evaluated the ability of #106 mAb to detect FMDV type A-specific SP Abs via SPCE using sera from cattle vaccinated with a bivalent vaccine containing inactivated FMDV type A (A22/Iraq/1964) and type O (O_1_ Manisa), following national routine vaccination programs in South Korea. Since the PrioCHECK SPCE is commonly used as a primary screening method worldwide, we simultaneously performed the commercial PrioCHECK SPCE for comparison. The PI value using HRP-conjugated #106 mAb was 89 ± 5%, while the PI value using the commercial mAb was 77 ± 13% (Figure 4B). Notably, SPCE using the HRP-conjugated #106 mAb showed PI values above 50 (cutoff value) in all the serum samples. However, the commercial SPCE exhibited PI values below 50 in one out of 12 samples under the same conditions. The results suggest that the use of the #106 mAb could improve the SPCE sensitivity for detecting FMDV type A SP Abs in the samples collected from susceptible animals.

## 3. Discussion

Diagnostic approaches for detecting Abs against FMDV are important to certify the health status of individual animals prior to import or export, to confirm suspected cases of FMD, to substantiate the absence or presence of FMDV infection, and to determine the efficacy of vaccines [11]. Since FMDV type A is one of the most common causes of FMD outbreaks [7], the vaccination against FMDV type A has been considered the best strategy for controlling FMD outbreaks. In this study, we developed #106 mAb specific to FMDV type A and focused on FMD type A in order to check the ability of our mAb to evaluate the FMD type A-specific vaccine efficacy (antibody production) using serum samples of vaccinated animals. The #106 mAb effectively bound to the inactivated FMDV type A. The #106 mAb had no cross-reactivity to inactivated FMDV types SAT 1, 2, and 3, and low reactivity to inactivated FMDV type O (O_1_ Manisa). Importantly, the use of HRP-conjugated #106 mAb in SPCE detected FMDV type A SP Abs in sera from FMD type A-vaccinated cattle more effectively than a commercial SPCE, suggesting that the #106 mAb could improve SPCE sensitivity.

In this study, using hybridoma systems, we selected the FMDV type A-specific #106 mAb, which had a higher antigen binding reactivity than those from other mAbs. Compared to a commercially available mAb, HRP-conjugated #106 mAb bound more efficiently to the inactivated FMDV type A. The comparison of the binding strength of the commercial mAb to our #106 mAb is limited as the antigenic origin of the commercial mAb is not known. The robust antigen binding capacity of #106 mAb could be advantageous to apply in various approaches, including mAb-based ELISA, fluorescent immunohistochemistry, or flow cytometry to detect the FMDV type A antigen. Because FMDV has seven different serotypes with high homology in amino acid sequences [18], cross-reactivity of the FMDV mAb is undesirable in terms of the correct diagnosis of certain FMDV subtypes. Here, we evaluated the cross-reactivity of the #106 mAb to FMDV types SAT 1, 2, and 3, which caused recent outbreaks in Africa [20,21,22] and FMDV type O, one of the widely distributed serotypes [7]. Fortunately, the #106 mAb had no cross-reactivities to the inactivated FMDV serotypes SAT 1, SAT 2, and SAT 3. The #106 mAb showed low reactivity to the inactivated FMDV type O (O_1_ Manisa). This reactivity could result from the high amino acid sequence identity (91.2%) between the serotypes A22/Iraq/1964 and O_1_ Manisa [18]. The commercial mAb had no cross-reactivity against FMDV type O, and SAT 1, 2, and 3, but #106 mAb showed more sensitivity to FMDV type A than the commercial mAb. A further study on cross-reactivity to other serotypes, including FMDV types Asia 1 and C, should be performed to fully investigate the specificity of this mAb. Also, the strength of the mAb can be improved by conjugation with various materials such as nanoparticles or by molecular engineering such as affinity maturation, specificity modulation, or stability engineering of the mAb as previous studies described [23,24,25].

The control and prevention of FMD are mainly based on vaccinations [26], and inactivated FMD vaccines are commonly used to combat the disease [27]. Because animals immunized with inactivated FMD vaccines produce Abs specific to the SPs [12], the diagnosis of SP Abs is considered the best way to evaluate the vaccine efficacy and is currently performed by three internationally accepted tests, including VNT, LPBE, and SPCE [11]. Among these tests, SPCE is widely used for primary screening to determine SP Abs in field samples due to the time and labor consumption of VNTs as well as the lack of antigen stability of LPBE. For the accurate diagnosis of FMD, SPCE for FMD is currently also needed to improve sensitivity and specificity [28]. Many efforts have been made to develop mAbs against SP Abs for FMDV type A [29,30,31], but it has been required to develop a FMDV type A SPCE with higher sensitivity and specificity until now. Herein, we demonstrated that our generated #106 mAb was more effective at detecting FMDV type A-specific SP Abs in the sera collected from the FMD-vaccinated cattle than a commercial SPCE. Notably, the use of our mAb in SPCE was able to detect FMDV type A SP Abs in all tested samples. However, the commercial SPCE failed to detect the Abs in one sample out of the 12 tested samples. The false negative diagnosis might result in not only the inaccurate investigation of the vaccine efficacy but also the misinterpretation of the infection status of animals before international trade, resulting in facilitating the spread of the disease. Although the high binding reactivity of the #106 mAb to FMDV type A can make false positive results, there is very low chance to cause problems because of false positives if our mAb is only used for evaluating FMDV type A vaccine efficacy. Notably, #106 mAb conjugate also showed a capability of reducing a false negative result when using PrioCHECK (Figure 4). Given the difficulty in obtaining the field samples until now, further study should be performed using large numbers of samples as well as diverse samples from other susceptible animal species, such as pigs and goats, to validate the broad use of our #106 mAb.

## 4. Materials and Methods

### 4.1. Mouse Immunization and Hybridoma Preparation

Six- to eight-week-old female BALB/c mice were purchased from Orient Bio (Gyeonggi-do, Korea) and housed in a specific pathogen-free facility at the Korea Research Institute of Bioscience and Biotechnology (KRIBB). Handling of mice and experimental procedures were reviewed and approved by the Institutional Animal Care and Use Committee of the KRIBB and were performed according to the Guidelines for Animal Experiments of the KRIBB (Approve number: KRIBB-AEC-17191). Mice were immunized thrice with 10 μg of inactivated FMDV type A (A22/Iraq/1964, Pirbright Institute, Pirbright, UK) mixed with the TiterMax Gold adjuvant (Sigma-Aldrich, St. Louis, MO, USA) by footpad injection on days 0, 14, and 28. Two weeks after each injection, sera from the immunized mice were collected. On day 42 after the initial administration, cells were isolated from the popliteal lymph nodes of the immunized mice and fused with myeloma FO cells (ATCC CRL1646) as previously described [32] to obtain hybridomas that produced anti-FMDV type A Abs.

### 4.2. Isotype Identification

The isotype of the mAbs was identified with an immunoglobulin isotyping kit (Roche Diagnostic, Mannheim, Germany). Briefly, mAbs were diluted in PBS, pH 7.2 at a concentration of 1 µg/mL. The diluted mAbs (150 µL) were added into the development tubes. The tubes were incubated at RT for 30 seconds and then vortexed to completely suspend the colored latex. Isotyping strips were placed in each development tube, and the results were interpreted after 10 minutes.

### 4.3. Conjugation of mAb with HRP

The mAb was conjugated with HRP using an HRP conjugation kit (Abcam, MA, USA) according to the manufacturer’s instructions. Briefly, the modifier reagent was added to the mAb (v:v = 1:10) and gently mixed. The mixture was then added to the lyophilized HRP and incubated at RT for 3 h. The reaction was stopped using the quencher reagent (v:v = 1:10) and stored at 4 °C until further use.

### 4.4. ELISA

MaxiSorp 96-well plates (Thermo Scientific, Roskilde, Denmark) were coated with 5 μg/mL inactivated FMDV type A (A22/Iraq/1964) in PBS overnight at 4 °C. Wells coated with 5 μg/mL BSA were used as negative controls. The plates were blocked with 5% skim milk in PBS for 2 h at 37 °C and washed with PBST. The plates were then incubated with the culture supernatants from the clones, and various concentrations of purified mAbs, HRP-conjugated mAb, or PrioCHECK conjugate (HRP-conjugated mAb specific to FMDV type A included in a PrioCHECK® FMDV Type A Kit (Prionics AG, Schlieren-Zurich, Switzerland) for 2 h at 37 °C. In the case of incubating with the culture supernatants of the clones and the purified mAbs, the plates were further incubated with HRP-conjugated anti-mouse IgG (1:5000) (Cell Signaling Technologies, Danvers, MA, USA) for 1 h at 37 °C. In separate experiments, MaxiSorp 96-well plates were coated with various concentrations of inactivated FMDV type A (A22/Iraq/1964) or BSA (negative controls) in PBS overnight at 4 °C. After blocking, the plates were incubated with either purified mAbs for 2 h at 37 °C, followed by incubation with HRP-conjugated anti-mouse IgG (1:5000) for 1 h at 37 °C or HRP-conjugated mAb and PrioCHECK conjugate for 2 h at 37 °C. Following PBST washing, the reactions were developed with the chromogenic tetramethylbenzidine (TMB) substrate (BD Biosciences, San Diego, CA, USA) and terminated with 2N H_2_SO_4_. Optical densities (ODs) were measured at 450 nm using a Versamax microplate reader (Molecular Devices, San Francisco, CA, USA).

In cross-reactivity experiments, MaxiSorp 96-well plates were coated with 1:500 dilution of virus culture supernatants of FMDV types SAT 1, 2, and 3 (inactivated) and 5 μg/mL inactivated FMDV types A (A22/Iraq/1964) or O (O_1_ Manisa; Pirbright) overnight at 4 °C. Wells coated with 5 μg/mL BSA were used as negative controls. After blocking, the plates were incubated with various concentrations of the purified mAb for 2 h at 37 °C, followed by HRP-conjugated anti-mouse IgG (1:5000) for 1 h at 37 °C, or various concentrations of HRP-conjugated mAbs or PrioCHECK conjugate for 2 h at 37 °C. In separate experiments, MaxiSorp 96-well plates were coated with various concentrations of virus culture supernatants of FMDV types SAT 1, 2, and 3 (1:12,800–1:200, inactivated), inactivated FMDV type O (O_1_ Manisa, 0.08–5 μg/mL; Pirbright), or BSA (0.08–5 μg/mL, negative controls) in PBS overnight at 4 °C. The FMD viruses SAT1/BOT/1/68, SAT2/ZIM/5/81 and SAT3/SIM/4/81 were used for antigen preparation in a baby hamster kidney 21-cell line. Twenty-four hours after virus infection, the viruses were inactivated by 0.003 N of binary ethylenimine for 24 h and concentrated with polyethylene glycol 6000 (Sigma-Aldrich). The virus was layered on 15%–45% sucrose density gradients and centrifuged. After the ultracentrifuge tube was punctured, 1 mL fractions were collected and then used for the experiments. After blocking, the plates were incubated with either 30 μg/mL purified mAb for 2 h at 37 °C, followed by incubation with HRP-conjugated anti-mouse IgG (1:5000) for 1 h at 37 °C, or 3 μg/mL HRP-conjugated mAb, or PrioCHECK conjugate (diluted according to the manufacturer’s instructions) for 2 h at 37 °C. After PBST washing, the reactions were developed with TMB and terminated with 2 N H_2_SO_4_. OD values were measured at 450 nm using the Versamax microplate reader.

### 4.5. Cattle Sample Collection

According to the national policy, routine vaccination programs against the FMDV types O and A have been performed in cattle in South Korea. Blood samples were collected from FMDV (A22/Iraq/1964 and O_1_ Manisa)-vaccinated cattle to slaughter at slaughterhouses, and serum samples were obtained with support from the Animal and Plant Quarantine Agency (Gimcheon, South Korea).

### 4.6. SPCE

Levels of Abs against SPs of FMDV type A in serum samples were determined by SPCE using HRP-conjugated mAb or commercial SPCE (PrioCHECK® FMDV Type A; Prionics AG, Schlieren-Zurich, Switzerland). The assays were performed according to the manufacturer’s instructions. Briefly, PrioCHECK plates coated with FMDV type A antigens were incubated with 1:5 dilutions of FMDV type A reference strong positive or negative sera (contained in the commercial SPCE kit) or serum samples from vaccinated cattle for 1 h at RT. After washing, the PrioCHECK conjugate (diluted following the manufacturer’s instructions) or 0.3 μg/mL of HRP-conjugated mAb was added to the plates and incubated for 1 h at RT. The plates were washed, and the TMB substrate was added. The reaction was stopped with 2 N H_2_SO_4_. OD values were measured at 450 nm using the Versamax microplate reader. The OD at 450 nm (OD 450 nm) of all samples were expressed as percentage PIs relative to the maximum of OD 450 nm. FMDV type A SP Abs were considered to be absent and present in the samples if the PI was below 50% and equal to or above 50%, respectively.

### 4.7. Statistical Analysis

Data are presented as the means ± standard deviation (SD) and represent at least two independent experiments. Statistical differences between the two groups or multiple groups were assessed using two-tailed Student’s *t*-test or one-way ANOVA followed by Bonferroni correction (ANOVA/Bonferroni), respectively. *p* values less than 0.05 (*p* < 0.05) were considered statistically significant. All analyses were performed using GraphPad PRISM software.

## 5. Conclusions

In conclusion, we herein report the development and application of a newly generated #106 mAb against SPs of FMDV type A. Our mAb showed high binding reactivity to inactivated FMDV type A (A22/Iraq/1964), low cross-reactivity to inactivated FMDV type O (O_1_ Manisa), and no cross-reactivity to inactivated FMDV types SAT 1, 2, and 3. Importantly, the SPCE using HRP-conjugated #106 mAb increased the sensitivity for detecting FMDV type A-specific Abs in the sera from FMDV type A-vaccinated cattle compared to a commercial SPCE. Therefore, the diagnostic application of our mAb might be useful for improving current SPCEs for determining the vaccine efficacy of FMDV type A.

## Figures and Tables

**Figure 1 pathogens-08-00301-f001:**
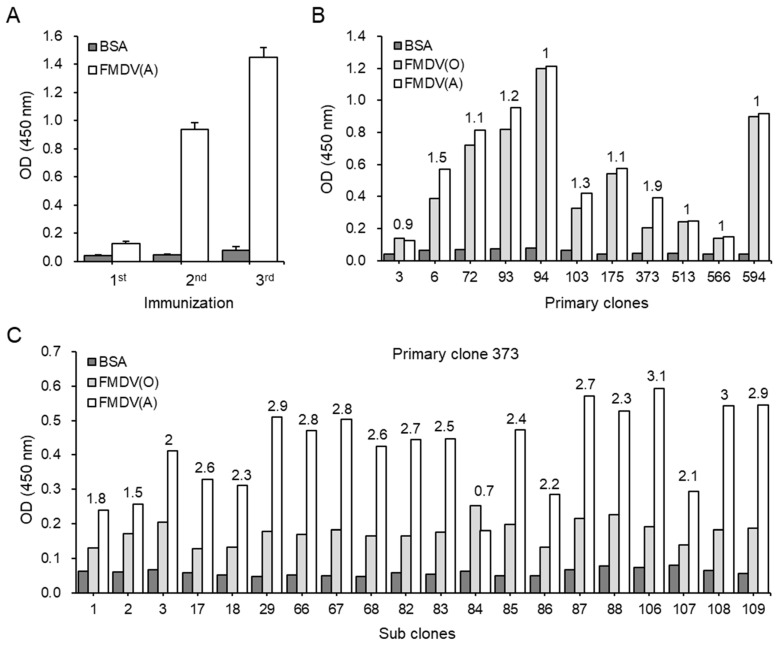
Production of anti-FMDV type A mAb-secreting hybridomas. BALB/c mice were immunized thrice with 10 μg of inactivated FMDV type A (A22/Iraq/1964) mixed with the TiterMax Gold adjuvant by footpad injection on days 0, 14, and 28. Levels of FMDV type A-specific Abs were measured in serum samples collected from the vaccinated mice two weeks after each immunization (**A**), in culture supernatants of primary clones (**B**), and in culture supernatants of subclones (**C**) by ELISA. Corrected OD values are shown. The numbers in B and C indicate fold differences in OD values of Abs reacted to inactivated FMDV type A compared to inactivated FMDV type O.

**Figure 2 pathogens-08-00301-f002:**
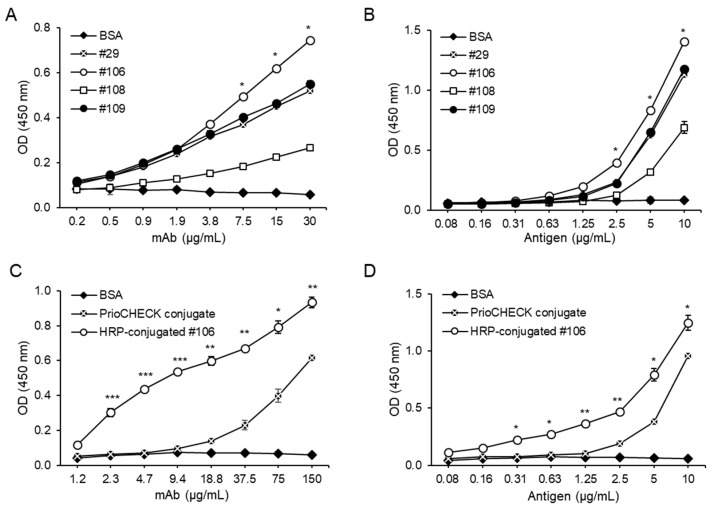
Binding reactivity of the mAbs against FMDV type A. (**A**,**C**) ELISA plates were coated with 5 μg/mL BSA (negative control) or inactivated FMDV type A (A22/Iraq/1964) overnight at 4 °C. After washing and blocking with 5% skim milk in phosphate buffer saline (PBS), the plates were incubated with 2-fold serial dilutions of either mAbs, followed by incubation with HRP-conjugated anti-mouse IgG (**A**) or HRP-conjugated #106 mAb and PrioCHECK conjugate (**C**). (**B**,**D**) Two-fold serial dilutions of BSA (negative control) or inactivated FMDV type A (A22/Iraq/1964) were coated overnight onto ELISA plates. After washing with 0.05% Tween-20 in PBS (PBST), the plates were blocked with 5% skim milk in PBS and then incubated with 30 μg/mL mAbs, followed by HRP-conjugated anti-mouse IgG (**B**) or 150 μg/mL HRP-conjugated #106 mAb and 150 μg/mL PrioCHECK conjugate (**D**). Corrected OD values are shown. The data are presented as the mean ± SD. Significant differences were analyzed by one-way ANOVA, * *p* < 0.05, ** *p* < 0.01, and *** *p* <0.001.

**Figure 3 pathogens-08-00301-f003:**
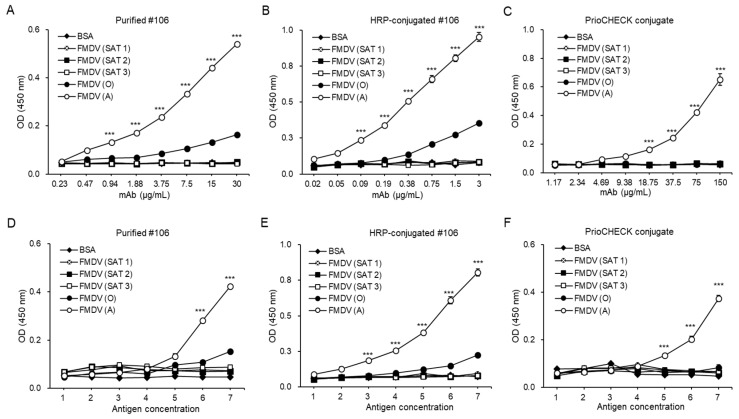
Specificity of the #106 mAb against different FMDV serotypes. (**A**–**C**) ELISA plates were coated overnight with 5 μg/mL BSA (negative control), 1:500 dilutions of virus culture supernatants of FMDV types SAT 1, 2, or 3 (inactivated), or 5 μg/mL of inactivated FMDV types A (A22/Iraq/1964), or O (O_1_ Manisa). After washing with PBST and blocking with 5% skim milk in PBS, the plates were incubated with 2-fold serial dilutions of either purified #106 mAb, followed by HRP-conjugated anti-mouse IgG (**A**), HRP-conjugated #106 mAb (**B**), or PrioCHECK conjugate (**C**). (**D**–**F**) Two-fold serial dilutions of BSA (0.08–5 μg/mL, negative control), virus culture supernatants of FMDV SAT 1, 2, and 3 (1:12800–1:200, inactivated), or inactivated FMDV type O and A (0.08–5 μg/mL) were coated overnight onto ELISA plates. After washing with PBST, the plates were blocked with 5% skim milk in PBS and then incubated with 30 μg/mL purified #106 mAb, followed by HRP-conjugated anti-mouse IgG (**D**), 3 μg/mL HRP-conjugated #106 mAb (**E**), or a predetermined dilution of PrioCHECK conjugate (**F**). Corrected OD values are shown. The data are presented as the mean ± SD. Significant differences were analyzed by one-way ANOVA, *** *p* < 0.001.

**Figure 4 pathogens-08-00301-f004:**
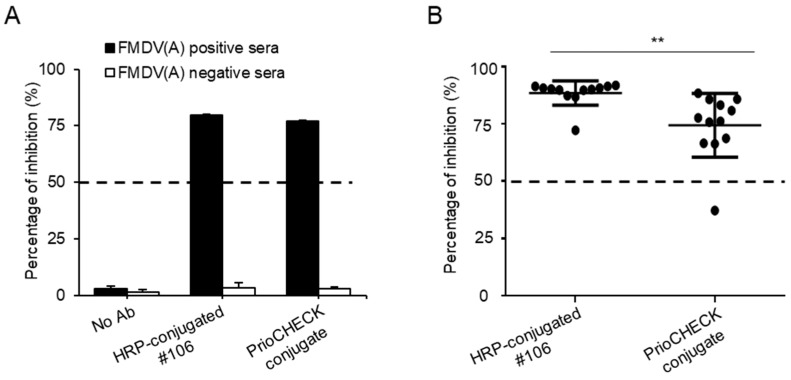
Detection of FMDV type A-specific structural protein (SP) Abs using the #106 mAb in SPCE. (**A**) Inactivated FMDV type A-coated plates were incubated with FMDV type A reference positive or negative sera. After washing, the plates were further incubated with assay buffer only (marked as “No Ab”) for a negative control, 0.3 μg/mL HRP-conjugated #106 mAb, or a predetermined dilution of PrioCHECK conjugate for 1 h at RT. Data shown are the mean values ± SD of duplicate samples and are representative of three similar independent experiments. (**B**) FMDV type A-coated plates were incubated with serum samples from FMDV type A-vaccinated cattle (*n* = 12) followed by incubation with HRP-conjugated #106 mAb or PrioCHECK conjugate for 1 h at RT. Data are presented as the mean ± SD of 12 samples and are presentative of two similar independent experiments. Significant differences were analyzed by a two-tailed *t*-test, ** *p* < 0.01.

**Table 1 pathogens-08-00301-t001:** Isotypes and EC_50_ values of the mAbs.

MAb	Isotype	Light chain	EC_50_ against Inactivated FMDV Type A (μg/mL)
#29	IgG2b	κ	12
#106	IgG2b	κ	5
#108	IgG2b	κ	124
#109	IgG2b	κ	10
HRP-conjugated #106	IgG2b	κ	5
PrioCHECK conjugate	IgG1	κ	69

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
