# Peer review of "Development of Monoclonal Antibody Specific to Foot-and-Mouth Disease Virus Type A for Serodiagnosis"

_pathogens, 2019, doi:10.3390/pathogens8040301_

Round 1

Reviewer 1 Report

This manuscript describes the generation of monoclonal antibodies (mAbs) specific to Foot and Mouth Virus (FMDV) type A, its characterization, and the selection of the most reactive one for developing a solid phase competitive (SPC) ELISA for detection of Abs against structural proteins (SP) of FMDV, type A-specific, in infected and vaccinated cattle. The authors propose that the in-house SPC-ELISA type A here described, might be useful for lowering cost of studies for vaccine efficacy and for diagnostic locally, where type A is the common FMDV type infecting animals.

I have some concerns about the interpretation of the results and the lack of data supporting the conclusions, and do recommend adding information of the kit performance and substantial changes in introduction and discussion. Here are some comments for the authors:

High binding reactivity to FMDV type A makes test more sensitive but is not always the best approach for herd testing given the high economic impact that may have the declaration of false positive animals. Anyway, to restrict detection to just one serotype is not a good policy for diagnosis ever. It would be useful to type positive FMD cases only, although would not discard the presence of other serotypes simultaneously. The #106 mAb showed very little but some reactivity to inactivated FMDV type South African territories 1, 2, and 3 and low reactivity to inactivated FMDV type O (O1 Manisa) but they did not compared to the crossreactivity of the commercial SPCE, and the number of samples was very limited both in number and in serotype These results suggest that the newly developed FMDV type A-specific mAb might be useful for lowering cost of diagnostic using an in-house kit instead of the commercial one in studies for vaccine efficacy and for diagnostic of A infected animals in scenarios without vaccination. However, in diagnostic for control and eradication while vaccinating detection of SP is not very useful since cannot differentiate vaccinated from infected, and for detection of infected animals during no vaccination it would be more informative to not be restricted to a single serotype.

The number of repeats to get the average and SD values are not specified (mean of X repeats) including the comparison of kits performance against 12 cattle sera: is the SD a sumatory of the 12 samples (thus the Priocheck having a negative result presents larger SD) or is the mean SD sample by sample. Does not make sense. Fig 4 is difficult to interpret since both conjugates behave so similarly in part A (please use only one graphic, repeating the graphic of the negative sera only is just confusing), but the PI of each individual animal is very different in dispersion in graphic B.

It is important that the manuscript provides the follow message:

to confirm FMDV circulation a universal detection like 3ABC is better: If vaccination is going on, the use is very limited because detection of SP do not allow differentiation between vaccinated and infected. Additionally, the test is restricted to detecting serotype A. Hence, the only advantage is for confirmation and serotyping of known FMD vaccinated, a positive result in animals vaccinated would not eliminate the possible circulation of a different serotype. This kit would be NOT USEFUL for this purpose if vaccination has been stopped the test will be able to type the SP antibodies against A exclusively, not giving free of other FMDV types: This kit would be NOT USEFUL for this purpose to determine FMD vaccine efficacy (only against A serotype):This kit would be USEFUL for this purpose.

Reviewer 2 Report

The authors cloned mAbs against FMD type A from vaccinated mice and compared them to the available commercial Ab.

Discussion:

      Discussion need to be improved to contain more information about the commercial Ab and how was it made and  to give better idea to the reader. 
How can the authors improve the strength of the mAb in the future. 
Can authors clone mAbs from infected cows?

Reviewer 3 Report

This manuscript by Nguyen and colleagues characterizes a novel monoclonal antibodies again a common type of foot-and-mouth disease virus (FDMV). FMDV is a significant health problem with ongoing epidemics worldwide. Additional diagnostic tools are necessary for rapid detection and accurate characterization of the FDMV type, in particular as vaccines against several FDMV types are available. In particular, there is a need for better antibodies for solid-phase competitive ELISA (SPCE) for immune monitoring following vaccination against FMD. The antibody with the highest sensitivity was then further compared against a commercial antibody for sensitivity but not specificity. This antibody could thus be a better candidate for immune testing in vaccinated animal. There are some points that should be clarified and addressed:

The terms “very little” and “weak” that are used in the statement that “the #106 antibody had very little cross-reactivity against inactivated FMDV types SAT 1, 2 and 3 as well as weak cross-reactivity to an inactivated FMDV type O” should be better justified or quantified.

Specificity was compared against the A antigen from the same A22/Iraq/1964 strain that was used to immunize the mice to generate the antibodies in the first place. The comparison to the commercial antibody would be more meaningful if it was known what FMDV A strain was used to generate this antibody.

The specificity of the #106 antibody was evaluated against four other FMDV types (O, Sat1,2 and 3) in Figure 3, but this was not compared to the commercial antibody.

The percentage of inhibition value was slightly higher compared to the commercial antibody in an SPCE with serum from immunized cattle, the main advantage of #106 seems to be that one out of twelve samples were missed with the commercial antibody. This seems overall marginal and should be acknowledged.

Of all seven FMDV types only five were tested for specificity. The authors acknowledge the need to also test cross reactivity against the other two types (C and Asia).

Round 2

Reviewer 1 Report

I appreciate the changes introduced in the manuscript

Reviewer 3 Report

I would suggest to add a statement that the comparison of the binding strength of the new antibody to the commercial antibody is limited as the antigenic origin of the commercial antibody is not known. 
